# Multifaceted Roles of the KEAP1–NRF2 System in Cancer and Inflammatory Disease Milieu

**DOI:** 10.3390/antiox11030538

**Published:** 2022-03-11

**Authors:** Harit Panda, Huaichun Wen, Mikiko Suzuki, Masayuki Yamamoto

**Affiliations:** 1Department of Medical Biochemistry, Tohoku University Graduate School of Medicine, Sendai 980-8575, Japan; harit.panda.r5@dc.tohoku.ac.jp (H.P.); wen.huaichun.s7@dc.tohoku.ac.jp (H.W.); 2Center for Radioisotope Sciences, Tohoku University Graduate School of Medicine, Sendai 980-8575, Japan; suzukimikiko@med.tohoku.ac.jp

**Keywords:** KEAP1, NRF2, NRF2-addicted/activated cancer, inflammation, sickle cell disease, cancer immunity

## Abstract

In a multicellular environment, many different types of cells interact with each other. The KEAP1–NRF2 system defends against electrophilic and oxidative stresses in various types of cells. However, the KEAP1–NRF2 system also regulates the expression of genes involved in cell proliferation and inflammation, indicating that the system plays cell type-specific roles. In this review, we introduce the multifarious roles of the KEAP1–NRF2 system in various types of cells, especially focusing on cancer and inflammatory diseases. Cancer cells frequently hijack the KEAP1–NRF2 system, and NRF2 activation confers cancer cells with a proliferative advantage and therapeutic resistance. In contrast, the activation of NRF2 in immune cells, especially in myeloid cells, suppresses tumor development. In chronic inflammatory diseases, such as sickle cell disease, NRF2 activation in myeloid and endothelial cells represses the expression of proinflammatory cytokine and adherent molecule genes, mitigating inflammation and organ damage. Based on these cell-specific roles played by the KEAP1–NRF2 system, NRF2 inducers have been utilized for the treatment of inflammatory diseases. In addition, the use of NRF2 inducers and/or inhibitors with canonical antineoplastic drugs is an emerging approach to cancer treatment.

## 1. Introduction—The Physiological Roles of the KEAP1–NRF2 Axis

The human body mitigates a plethora of chemical stresses that it encounters daily. These stressors range from extrinsic environmental toxins, including air and water pollutants released from factories and industries, micro/nanoplastic particles and ultraviolet radiation, to intrinsic factors, such as reactive oxygen species (ROS) and proinflammatory cytokines. Our bodies are equipped with various cytoprotective machineries to combat the effects of these stressors and maintain redox homeostasis. Of the body’s multitude mitigation strategies, the KEAP1–NRF2 system is recognized as one of the supreme contributors to the defense against such environmental and intrinsic insults.

The transcription factor nuclear factor erythroid 2-like 2 (NRF2, encoded by *nfe2l2*) is a master regulator of the cellular response against oxidative and electrophilic stresses and plays a vital role in protection against both endogenous and exogenous insults [1,2]. Kelch-like ECH-associated protein 1 (KEAP1) is a cysteine thiol-enriched molecule that acts as a stress sensor for NRF2. KEAP1 is an adaptor molecule of a ubiquitin E3 ligase complex and forms a complex with CULLIN3 (Cul3) and other factors [3].

In the cytoplasm under quiescent conditions, NRF2 is polyubiquitinated by the KEAP1–CUL3 ubiquitin E3 ligase complex, which leads to rapid degradation of NRF2 through the proteasome system. In unstressed cells, NRF2 is constantly degraded. Hence, although NRF2 is synthesized in relative abundance, NRF2 is maintained at a low level within the cells [2]. However, upon exposure to oxidative or electrophilic stress, the cysteine residues in KEAP1 are modified, and KEAP1 loses ubiquitin ligase activity, allowing NRF2 to migrate into the nucleus (Figure 1). As a result, NRF2 rapidly accumulates in the nucleus. NRF2 then heterodimerizes with small MAF (sMAF) family proteins, and NRF2–sMAF heterodimers bind to cis-regulatory elements called antioxidant-responsive elements (AREs) [3] or electrophile-responsive elements (EpREs) [4]. AREs and EpREs are collectively called CNC–sMaf binding elements (CsMBEs) [5].

The binding of an NRF2–sMAF heterodimer complex to a CsMBE region results in robust activation of target genes. Canonical NRF2 target genes include NADPH: quinone oxidoreductase 1 (encoded by *Nqo1*), glutamate cysteine ligase catalytic and modifier subunits (*Gclc* and *Gclm*), and ROS elimination antioxidant enzymes, including heme oxygenase 1 (*Hmox1*) and peroxiredoxin 1 (*Prdx1*) [5,6,7,8,9,10]. The list of NRF2 target genes has been expanding because of their identification in various medical, biological and genome analyses, especially in chromatin immunoprecipitation-sequencing (ChIP-seq) analysis, which has been an important advancement in the study of the KEAP1–NRF2 system [7].

Extensive studies utilizing NRF2-null and KEAP1-deficient animals have been carried out in many laboratories and have demonstrated the critical cytoprotective role NRF2 plays in cells challenged by various extrinsic insults. For example, when NRF2-knockout mice and rats grow normally, they are more sensitive to oxidative and xenobiotic stress than wild-type animals [1,11]. For instance, it has been shown that NRF2-deficient mice are more susceptible to cigarette smoke-induced emphysema [12], hyperoxic lung injury [13], and pulmonary fibrosis induced by bleomycin [14].

On the other hand, KEAP1-knockdown mice, by virtue of their high NRF2 levels, are resistant to the aforementioned stresses. Notably, hepatocyte-specific deletion of the KEAP1 gene conferred potent resistance against acute drug toxicity [15]. Therefore, pharmacological activation of NRF2 is expected to confer similar protection, and NRF2 inducers are under development, with some currently available for use in laboratory experiments. In fact, acetaminophen hepatotoxicity [16] and cigarette smoke-induced emphysema have been prevented by treatment with 2-cyano-3,12 dioxooleana-1,9 diene-28-imidazolide (CDDO-Im) [17].

Hence, the KEAP1–NRF2 system is an attractive target for therapeutic interventions in various oxidative stress-linked maladies. In fact, we have shown through animal studies that genetic and pharmacological activation of NRF2 ameliorates many pathological conditions, such as multiple sclerosis [18], chronic obstructive pulmonary disease (COPD) [19], Alzheimer’s diseases [20], diabetes mellitus [21], and sickle cell disease (SCD) [22,23]. The NRF2 inducer dimethyl fumarate (DMF, Tecfidera^®^) has been approved by the US Food and Drug Administration (FDA) for the treatment of multiple sclerosis. Mechanistic support for the DMF-mediated induction of NRF2 activation has been provided by studies on renal cell carcinoma [24,25]. In addition, bardoxolone (CDDO-Me), an oleanane triterpenoid class NRF2 inducer, is currently in a phase 3 clinical trial in Japan (Ayame trial) for the treatment of diabetic nephropathy [26].

## 2. Structural Basis of KEAP1 and NRF2 Functions

### 2.1. KEAP1 Senses Electrophiles and ROS

KEAP1 is composed of 685 amino acid residues and contains many reactive cysteine residues: 25 in mice and 27 in humans. Extensive molecular dissection of KEAP1 has been performed, and the results have shown that KEAP1 comprises three functional domains (Figure 2A): the N-terminal Broad complex, Tramtrack, and Bric-a-Bric (BTB) domains, central intervening region (IVR), and C-terminal double glycine repeat and C-terminal (DC) domain. The DC domain consists of a carboxyl-terminal region (CTR) and a double glycine repeat (DGR) domain [2].

Of these domains and regions, the BTB domain is critical for Keap1 homodimerization and binding to CUL3 [3,27,28]. A critical finding in biochemical studies of KEAP1 has shown that reactive cysteine residues in KEAP1 bind covalently to electrophiles, suggesting that they act as sensors for toxic chemicals [29,30,31]. The significance of cysteine residues as sensors has been shown in various in vitro [32,33,34], zebrafish [33], and mouse [35,36,37] experiments. These studies have revealed three critical cysteine residues in KEAP1: Cys151 in the BTB domain and Cys273 and Cys288 in the IVR. Cysteine residues contain a thiol (-SH) group that has high affinity for electrophiles, and various established NRF2 inducers have been shown to interact with these three cysteine residues because of their electrophilic nature.

Based on the cysteine moiety with which these electrophiles and other non-electrophilic inducers interact, electrophiles are divided into five groups (Figure 2B). Class I consists of Cys151 preferred electrophiles, which include diethylmaleate (DEM), DMF, sulforaphane (SFN), CDDO-Im, and nitric oxide (NO). Class II electrophiles have affinity for all three cysteines, Cys151, Cys273, and Cys288, and 9-nitro-cotadec-9-enoic (OA-NO_2_), 4-hydroxynonenal (4-HNE), and NaAsO_2_ belong to this group. Class III sensing is mediated solely by Cys288, and currently, only one electrophile, 15d-PGJ_2_, is in this class.

To find sensor cysteine residues that interact with ROS in addition to these three cysteines, which are crucial for sensing electrophiles, four cysteine residues, Cys226, Cys613, Cys622, and Cys624, have been recently discovered to be important for H_2_O_2_ sensing [38]. These four cysteine residues constitute the Class IV category. Notably, electrophiles and ROS are sensed by distinct sets of cysteine residues in KEAP1.

In addition to the cysteine residue sensors, or the cysteine codons, inhibition of the KEAP1–NRF2 protein–protein interaction (PPI) has been found to induce NRF2 activation. For instance, the autophagy chaperone molecule p62 directly binds to KEAP1, thereby competing with NRF2 [39]. Several small molecules, such as Cpd16, PRL295, and NG262, have also been found to inhibit the PPI of KEAP1 and NRF2 [40,41]. These observations indicate that the interface of the KEAP1–NRF2 interaction acts as a site where cellular signals are sensed [40,41].

### 2.2. NRF2 Is Composed of Six Functional Domains

The phylogenetically conserved structure of NRF2 in various species has six functional domains referred to as NRF2-ECH homology (Neh) domains, and the Neh1 to Neh6 domains have been identified (Figure 3A) [2]. The Neh1 domain harbors a Cap and collar (CNC) structure and a basic region-leucine zipper (bZIP) structure, which both form dimers with a sMAF and bind DNA [42]. On the other hand, the Neh4 and Neh5 domains are transactivation domains that mediate interactions with coactivators. Neh4 and Neh5 make major contributions to transcriptional activation through NRF2 by recruiting the histone acetyl-transferase the cAMP responsive element binding (CBP) protein [43] and mediator complex [44]. The Neh2 domain is in the N-terminal region and contains seven lysine residues in a centralized α-helix region, and these lysine residues are targets of ubiquitinase (Figure 3B). Neh2 harbors two distinct peptide motifs (DLG and ETGE) on both sides of the α-helix region, and these motifs interact with KEAP1 [2]. These two motifs individually bind to a specific binding pocket in the KEAP1 DC domain [45]. The classical DLG motif has been represented from amino acid Leu23 to Gly31, but later studies based on somatic mutation analyses revealed that the motif was much longer; the newly longer form of the motif was named DLGex (extended DLG) and extends from Met17 to Gln51 [46].

A thermodynamic study has revealed a critical difference between ETGE and DLGex motifs in their interactions with the DC domain of KEAP1 [46] (Figure 3B). Although KEAP1-DLGex binding is known to be driven by both enthalpy and entropy, KEAP1-ETGE binding is exclusively driven by enthalpy. Moreover, KEAP1-ETGE binding has been shown to be realized through a two-state binding reaction, leading to a much more stable confirmation than KEAP1-DLGex binding [46]. This difference explains the relatively high affinity of ETGE for the DC domain and forms the basis for a two-site binding/hinge-and-latch model showing the disruption of the KEAP1–NRF2 interaction with motifs behaving similar to a door hinge and latch [45,46,47].

Recent nuclear magnetic resonance (NMR) studies have suggested that the hinge-and-latch model is accurate, as Nrf2 activation by accumulated p62 (SQSTM1), an autophagy chaperone, is mediated through a hinge-and-latch mechanism. Similarly, the same study revealed that the KEAP1–NRF2 PPI inhibitors PRL295 and NG262 negate binding through the hinge-latch mechanism. These PPI inhibitors preferentially disrupt DLGex-KEAP1 binding owing to the lower affinity of DLGex for KEAP1. In contrast, electrophilic NRF2 inducers use distinct mechanisms to induce NRF2 activation. The aforementioned inhibitors are thiol-modifying chemicals that covalently bind to KEAP1 cysteine residues, causing structural changes in the KEAP1 protein to prevent NRF2 ubiquitination. Further studies are warranted to obtain a clearer picture of these processes [47].

The Neh6 degron is an additional NRF2 regulatory modality [48]. KEAP1-independent regulation through Neh6 is facilitated by the E3 ubiquitin ligase β-transducin repeat-containing protein (β-TrCP) and requires prior glycogen synthase kinase 3β (GSK-3β)-induced phosphorylation of the 335th and 338th serine residues in the Neh6 domain. The phosphorylation of NRF2 enables NRF2 to be recognized by the β-TrCP complex for ubiquitination and, ultimately, the degradation of NRF2 by the β-TrCP-CUL1 E3 ubiquitin ligase complex [49,50]. GSK-3β can be inhibited by phosphatidyl inositol 3 kinase (PI3K)-AKT mediated phosphorylation [51,52].

## 3. The KEAP1–NRF2 Axis in Cancer

### 3.1. NRF2 Is Constitutively Activated in NRF2-Addicted Cancer

Although NRF2 normally helps protect cells against oxidative stresses by coordinately activating genes encoding detoxifying and antioxidant proteins, constitutive NRF2 accumulation confers cancer cells with chemo/radiotherapy resistance [53] and provokes malignant growth. The advantages conferred to these cancers to withstand the effects of chemotherapeutic drugs, such as etoposide and carboplatin, have been previously reported [54]. These kinds of cancers are referred to as NRF2-addicted/NRF2-activated cancer. Having undergone metabolic reprogramming, resulting in extreme proliferation [10] and resistance to anticancer therapeutics, NRF2-addicted cancer cells are difficult to manage in the clinic. Hence, NRF2-addicted tumors are associated with poor prognosis for patients.

Somatic mutations in the *KEAP1* and *NRF2* genes are frequently observed in various cancers. The earliest reports of aberrant activation of NRF2 through somatic mutations in KEAP1 [55] and NRF2 [56,57] were based on human non-small-cell lung carcinoma (NSCLC). Since then, many similar mutations have been found in various other cancers, such as breast [58], gall bladder [59], skin, and liver [55,57,60] cancers. Recently, next-generation sequencing (NGS) deep-sequencing data have also revealed recurrent KEAP1 and NRF2 mutations in solid tumors of the head and neck, upper digestive tract, and renal cell carcinoma [48,61]. In addition to mutations in KEAP1 and NRF2, missense mutations in Cul3 have been frequently observed in head and neck and papillary renal cell carcinoma [61,62].

A recent comprehensive genomic study of NSCLC patients elucidated important ancestry-based disparities in the genomes of East Asian and Caucasian groups. Intriguingly, the report highlighted that NSCLCs in Japanese individual showed fewer mutations compared to those in individuals of European descent. This variation in the mutation incidence included alterations in the *KEAP1* gene. Although somatic mutations in *KEAP1* was found to have occurred at a rate of 17.8% in NSCLCs in the European cohort, they were found in only 0.5% of all NSCLCs in the Japanese cohort (Figure 4). This difference was not found for the *NRF2* (*NFE2L2*) gene. Somatic mutations in the *NRF2* gene were found in 15.8% and 13.6% of European and Asian patients, respectively [63].

In NRF2, the DLG and ETGE motifs of the Neh2 degron exclusively bore the aforementioned mutations, and as noted above, these motifs are important for KEAP1–NRF2 binding. These findings strengthen the argument that the DLG and ETGE domains play important roles in NRF2 activation and support the previously described two-site binding/hinge and latch binding model of NRF2 stabilization [64]. In contrast, mutations in KEAP1 and CUL3 were spaced throughout the coding region [65]. Interestingly, mutations in NRF2, KEAP1, and CUL3 were mutually exclusive, suggesting that they all follow the same mechanism to provide a growth advantage to cancer cells, i.e., hyperactivation of NRF2 [66,67].

In addition to these mutant sequences, at least four pathways are involved in NRF2 activation in cancer cells [68]. The first pathway involves epigenetic silencing of *KEAP1* gene expression through DNA methylation of the *KEAP1* gene promoter. This modification results in NRF2 stabilization. Epigenetic abnormalities have been reported in cases of malignant gliomas and lung tumors [69,70]. Increased accumulation of p62/SQSTM1, hampering the KEAP1–NRF2 interaction, is a common characteristic in hepatocellular carcinoma [71,72]. This dysregulated protein accumulation is the second pathway critical for NRF2 activation independent of somatic mutations. The third pathway entails cysteine modification in KEAP1 through oncometabolites, such as fumarate, causing persistent NRF2 activation similar to that observed in smooth cell tumors [73]. Exon skipping in the *NRF2* gene has also been seen, wherein the NRF2 domain is deprived of the degrons required for KEAP1 interactions or those important in its ubiquitination [74]. In fact, a high incidence of *NRF2* mRNA devoid of exon 2 has been found in lung cancers. Exon skipping is the mechanism utilized in the fourth pathway, and there may be additional mechanisms. All these molecular events ultimately result in disruption of the KEAP1–NRF2 interaction, leading to NRF2 stabilization and persistent activation of its downstream genes, which confer protection to cancer cells from radio/chemotherapy and boost their expansion.

How does constitutive NRF2 activation bestow tumors with proliferative advantages through molecular reprogramming? The mechanism underlying this process has been extensively addressed in various studies [10,75,76]. NRF2 in proliferating cells has been shown to augment glutamine consumption and purine nucleotide synthesis, leading to tissue hypertrophy [10]. In fact, NRF2 has been shown to directly upregulate the expression of metabolic enzymes crucial in the pentose phosphate pathway and for NADPH production, such as glucose-6-phosphate dehydrogenase *(G6PD*), phosphogluconate dehydrogenase (*PGD*), transaldolase 1 (*TALDO1*), transketolase (*TKT*), and malic enzyme 1 (*ME1*) [10]. In agreement, a recent study has shown that mouse lung cancer cells with KEAP1 and KRAS co-mutations are dependent on glutaminolysis [75]. The key NRF2-dependent factor critical for this effect has been identified through CRISPR library screening to be the glutamine transporter *Slc1a5* [75].

Additionally, comprehensive metabolomic profiling of human lung cancer cells has shown that NRF2 also controls the serine biosynthesis metabolic program via activating transcription factor 4 (ATF4) and phosphoglycerate dehydrogenase activation [76]. Serine is subsequently used to supply substrates for glutathione and nucleotide production; these substrates and nucleotides are ultimately used in the regulation of the pentose phosphate pathway to supply ribose for nucleotide production. These findings suggest that multiple NRF2-regulated pathways coordinately contribute to the malignant development of tumors.

### 3.2. NRF2 Plays Diverse Roles in Different Cell Fractions in the Tumor Microenvironment

The pathology of cancers is dictated not only by tumor cells but also by other resident cells in the microenvironment. Variations in the expression levels of NRF2 in these tumors, as well as surrounding cells in the microenvironment exert distinctive effects on tumorigenesis, with either positive or negative implications in tumor progression. In this section, we describe several salient studies on NRF2 in different niches.

#### 3.2.1. NRF2-Null Host Cells Are Eliminated by Competitive Cell Selection during Chemical Carcinogenesis

Cell competition is an emerging topic of interest. One recent study pointed out that cellular NRF2 levels determined cell fate during chemical carcinogenesis in the esophageal epithelium [77]. Using tamoxifen-induced squamous epithelium-specific NRF2 conditional-knockout mice, the fate of NRF2-deficient epithelial cells in esophageal carcinogenesis was determined. When NRF2 was deleted in 50% of esophageal epithelial cells prior to carcinogen exposure to 4NQO, the NRF2-deficient cells were eliminated by competitive cellular selection (Figure 5A). Therefore, only NRF2-intact neighboring cells underwent carcinogenesis. On the other hand, when NRF2 was knocked out in 50% of the esophageal epithelial cells after 4NQO treatment, the resulting tumors comprised both NRF2-intact and NRF2-null cells. These results clearly show that the selective elimination of NRF2-null cells is a defense mechanism in esophageal epithelial tissue that prevents carcinogenesis.

#### 3.2.2. NRF2 Activation in Immune Cells Leads to Tumor Suppression

Recent progress in the pleiotropic functions of NRF2 revealed that NRF2 activation in host cells, especially in the host immune system, suppresses the aberrant growth of NRF2-activated cancer cells. In line with this finding, activation of NRF2 in the microenvironment has been shown to inhibit the progression of NRF2-activated malignant tumors [78,79,80].

In this regard, a question arises: how do the immune cells in the tumor microenvironment influence tumorigenesis? Myeloid-derived suppressor cells (MDSCs) are referred to as the queen bees of the tumor microenvironment because MDSCs play important roles in the tumor microenvironment. Although healthy people usually lack MDSCs, chronic inflammation that promotes the onset and development of tumors also stimulates the appearance and increases the number of MDSCs [81]. Circulating MDSCs are associated with low survival rates, and the risk of dying from cancer is almost two fold higher in patients with MDSCs than in patients without MDSCs [82]. Most importantly, MDSCs have been found to have suppressive effects on both innate and adaptive immunity, which facilitates an increase in tumor immune tolerance [83]. Intracellular ROS play pivotal roles in MDSC-mediated immunosuppression. ROS accumulation in MDSCs suppresses the activation and proliferation of CD8^+^ T cells by modifying the T-cell receptors on their surface, which disrupts their interaction with major histocompatibility complex class I molecules on antigen-presenting cells [84]. Notably, CD8^+^ T cells directly attack tumor cells and play a vital role in tumor suppression.

In a mouse model with systemic and myeloid cell specific NRF2 knocked out and transplanted 3 Lewis lung carcinoma (3LL) cells, higher ROS accumulation in the Mac1^+^Gr1^+^ MDSC cell fraction was observed [78,79]. These mice also displayed a higher rate of cancer incidence and spontaneous lung metastasis, clearly demonstrating that high NRF2 levels in the tumor microenvironment suppressed tumor cell development by reducing the ROS levels in MDSCs (Figure 5B). In contrast, a decrease in NRF2 levels led to the accumulation of ROS in MDSC cells, which, in turn, suppressed CD8^+^ T cells (innate immunity), generating a favorable environment for the homing of cancer cells. These observations indicate that NRF2 activation in host myeloid cells can be a therapeutic target in clinical settings.

The emergence of MDSCs is promoted by numerous factors, including interleukin-6 (IL-6). For instance, it has been shown that NRF2 interferes with the upregulated expression of proinflammatory cytokines, including IL-6, in macrophages and that this NRF2 function is independent of ROS levels [85]. Hence, it seems reasonable to speculate that NRF2 plays multiple roles in MDSC suppression; specifically, its effect is either ROS-independent and IL-6 repression-dependent or dependent on a reduction in ROS levels (Figure 5B). Additionally, restoration of CD8^+^ T-cell activity has been observed in a microenvironment with high levels of Nrf2, which displays lower levels of the immunosuppression marker programmed death receptor-1 (PD-1) and higher levels of the T-cell activation markers TNFα and IFNγ [80] (Figure 5B).

### 3.3. NRF2 Inhibitors in the Treatment of NRF2-Addicted Cancers

For the treatment of NRF2-addicted/activated cancers with high NRF2 dependency, the use of NRF2 inhibitors seems to be a promising approach. NRF2 inhibitors antagonize cancer growth and sensitize cancer cells to therapy. Germane to this hypothesis, the plant-based chemical compound brusatol has been shown to decrease NRF2 protein levels in tumors, making the tumors more sensitive to chemotherapy [86]. Halofuginone, a synthetic halogenate derivative of febrifugine, showed a similar impact on NRF2-addicted tumors by suppressing NRF2 accumulation [87]. Halofuginone inhibits prolyl-tRNA synthetase function leading to accumulation of uncharged tRNA. Increased halofuginone levels elicited an amino acid starvation response and repressed global protein synthesis. Since NRF2 is a very short-lived protein with a half-life before degradation of less than 20 min [88], blocking general protein synthesis significantly inhibited NRF2 synthesis [87].

In comparison to that on NRF2 inducers, however, research on NRF2 inhibitors is still in its infancy. Currently available NRF2 inhibitors exhibit systemic NRF2 inhibitory properties and subsequent off-target effects; e.g., systemic NRF2 inhibition reduces NRF2 levels not only in the tumor but also in the surrounding cells, including immune cells. In contrast to its effect on NRF2-addicted cancer cells, NRF2 is protects normal cells against oxidative and xenobiotic stress. In fact, increased NRF2 in immune cells also plays roles in tumor cessation or cancer immunity [78,79]. Therefore, it is reasonable to raise concerns over inhibiting NRF2 in normal cells, which might lead to cytotoxic effects.

One way to attenuate systemic NRF2 inhibition is to employ a drug delivery system based on the principle of the enhanced permeability and retention (EPR) effect [89]. Because of its abnormal growth, tumor tissue usually has an aberrant vascular architecture and gaps in the endothelial lining. Furthermore, most tumor tissues lack a lymphatic drainage system. These defects form the basis for the EPR effect. By incorporating an NRF2 inhibitor in a polymeric micelle which has high tumor specificity, NRF2 inhibitor’s effect can be localized to the cancerous tissue and limit its toxicity in other normal cells (Figure 6). An alternate way to attenuate off-target effects is to find a target that is exclusively expressed in NRF2-addicted cancer, such as the orphan nuclear receptor NR0B1, which has been shown to be expressed in non-lung cell carcinoma [90].

Recently, a new approach to the treatment of NRF2-addicted malignant cancers has been developed, and it is based on increasing NRF2 activity to transform chemotherapy prodrugs into truly toxic chemotherapy drugs. In this strategy, called synthetic lethality, chemotherapy drugs with mild efficacy are transformed into strongly active drugs by NRF2-targeted drug-metabolizing enzymes. For example, geldanamycin-derived HSP90 inhibitors can be transformed to be strongly active drugs in NRF2-activated cancer cells [91]. Intriguingly, a widely used chemotherapy drug, mitomycin C, becomes more toxic in cancer cells with aberrant NRF2 activation because mitomycin C is transformed into a more toxic form by the enzymes induced by NRF2 [92].

### 3.4. NRF2 Inducers for the Treatment of NRF2-Activated Cancers

Nrf2 activation in the microenvironment has been reported to increase the survival of mice with Nrf2-activated cancer. Additionally, Nrf2-activated immune cells suppress tumor burden and preinvasive lesion formation [80]. Hence, using Nrf2 inducers for the treatment of Nrf2-activated cancers is a plausible therapeutic strategy. In fact, administration of the NRF2 inducer CDDO-Im suppressed tumor growth and metastasis in a 3LL murine cancer xenograft model (Figure 6) [79]. We surmise that certain combinations of NRF2 inhibitors and NRF2 inducers may show good efficacy.

## 4. KEAP1–NRF2 Axis and Inflammation

### 4.1. NRF2 Activation Resolves Acute and Chronic Inflammation

Although inflammation is a protective immune response against various external as well as internal stimuli, prolonged chronic inflammation provokes detrimental effects. In addition to its roles in the oxidative stress response, NRF2 plays a vital role in anti-inflammatory activities. NRF2 activation protected mice challenged with lipopolysaccharide (LPS) from mortality, while LPS-induced acute inflammation made Nrf2-knockout mice vulnerable [93]. Nrf2-deficient mice exposed to cigarette smoke showed higher neutrophilic lung inflammation and higher susceptibility to emphysema than wild-type mice [19].

Notably, the endogenous NRF2 inducer 15-deoxy-Δ^12,14^-prostaglandin J_2_ (15d-PGJ_2_), one of the end products of the cyclooxygenase-2 (COX-2) pathway, plays an important role in the anti-inflammatory function of NRF2. Then, 15d-PGJ_2_ inhibits inflammation by downregulating the expression of proinflammatory transcription factors, such as NF-κB. Intriguingly, 15d-PGJ_2_ induces NRF2 activity by binding to Cys288 in KEAP1. In fact, the anti-inflammatory effect of 15d-PGJ_2_ was abrogated in Nrf2-deficient mice [94], supporting the notion that NRF2 counteracts or resolve acute inflammation. 

NRF2 also mitigates chronic inflammation, which has been established in various studies, such as in mouse model studies of multiple sclerosis and SCD. Tissue damage in multiple sclerosis is caused by chronic inflammation and oxidative stress. In vitro studies have demonstrated that DMF, an NRF2 inducer, increased murine neuron survival and protected human- or rodent-derived astrocytes from oxidative stress [95]. In vivo studies with experimental autoimmune encephalomyelitis (EAE) model mice, which serve as a murine model of multiple sclerosis, supported these observations [95]. Administration of an oral formulation of DMF enhanced the preservation of myelin, axons, and neurons in EAE mice [95]. In fact, DMF is now an FDA-approved drug (Tecfidera^®^) used to prevent relapse of multiple sclerosis. The role of NRF2 in mitigating another malady associated with chronic inflammation, SCD, is discussed in the following sections.

### 4.2. Molecular Basis of NRF2 Anti-Inflammatory Activity

Recent studies have revealed that NRF2 alleviates acute and chronic inflammation through two distinct mechanisms [85]. One mechanism is ROS-dependent (Figure 7, left panel). It has been established that NRF2 induces the expression of anti-oxidative genes and decreases cellular ROS levels [85]. ROS are important causes of tissue damage, and they can stimulate the progression of inflammation; however, NRF2 activation confers strong protection against the progression of inflammation by reducing the ROS level through the antioxidant proteins.

The other NRF2-based anti-inflammatory mechanism is ROS-independent. NRF2 represses the activation of proinflammatory cytokines, such as IL-6 and IL-1β, in inflammatory tissues. Through ChIP-seq and ChIP-qPCR analysis, NRF2 has been shown to bind to the regulatory regions of proinflammatory genes in macrophages and hence impede the recruitment of RNA polymerase II without affecting the NF-κB recruitment required for the expression of IL-6 and IL-1β genes [85] (Figure 7, right panel). This discovery is particularly interesting because until recently NRF2 was recognized as an activating transcription factor, but this study clearly revealed that NRF2 also has transcription-inhibiting function. Although the precise molecular mechanisms of NRF2 function, as well as the cell-type specificity of this regulatory effect remain to be clarified, the contribution of the ROS-independent pathway has been confirmed in analyses of NRF2 treatment of EAE model mice [85].

### 4.3. NRF2 Activation Attenuates SCD Pathology

SCD is an inherited hemoglobinopathy caused by the production of mutant hemoglobin S (HbS) [96]. SCD affects millions of people worldwide and is one of the main causes of morbidity and mortality globally. SCD is caused by a single amino acid substitution (a glutamic acid residue is replaced with a valine residue) at the sixth position in the β-globin gene [97]. Under low-oxygen conditions, HbS leads to the production of sickle-shaped erythrocytes that are vulnerable to hemolysis. This defect ultimately leads to intermittent vaso-occlusion, ischemia/reperfusion injury, and high oxidative stress [98]. Additionally, free heme and iron released from the hemolyzed red blood cells further increases vaso-occlusion and adds to the oxidative stress and inflammation in a patient with sickle cell disease [99,100].

It has been demonstrated that genetic and pharmacological activation of NRF2 ameliorates inflammation and organ damage in SCD [22]. By crossing SCD model mice harboring a mutant human β-globin gene (hβ^S/S^) [101] with *Keap1*-knockdown mice (*Keap1*^F/−^), compound hβ^S/S^::*Keap1*^F/−^ sickle mice that harbored systemic and constitutive activation of Nrf2 expression were obtained. Compared with the control hβ^S/S^::*Keap1*^+/+^ mice, the hβ^S/S^::*Keap1*^F/−^ mice presented with significantly reduced inflammation and lower levels of the proinflammatory cytokines IL-6, IL-1β, and IL-18 in lung tissue and vascular cell adhesion molecule (VCAM) and P-selectin in the aorta. Liver damage due to SCD was also dramatically reduced in the hβ^S/S^::*Keap1*^F/−^mice. Consistent with this observation, SCD::*Nrf2^−/−^* model mice showed decreased expression of antioxidant proteins, resulting in increased levels of ROS, proinflammatory cytokines and adhesion molecules compared to the control SCD::*Nrf2^+/+^* mice [102].

### 4.4. Multifarious Roles of NRF2 in Different Types of Cells in SCD

A key question remained, is systemic NRF2 activation important for NRF2-mediated attenuation of SCD pathology, or is it vital only in certain crucial cell lineages? To answer this question, a study with sickle murine model mice in which KEAP1 was specifically deleted in myeloid lineage cells (SCD::*Keap1^F/F^*::*LysM-Cre*) or endothelial cells (SCD::*Keap1^F/F^*::*Tie1-Cre*) has been carried out [103]. Among the many other cells that might participate in SCD pathology, endothelial cells, and myeloid lineage cells were selected for evaluation because they reside in proximal to areas with ischemic/reperfusion injury and heme metabolism, respectively.

Keap1 deletion in myeloid cells (SCD::*Keap1^F/F^*::*LysM-Cre*) was found to ameliorate inflammation, as evidenced by the lower white blood cell (WBC) count, reduced mRNA expression of the cytokines TNFα and IL-1β, and lower congestion, edema and inflammatory cell infiltration in the lung cross-sections, which were evaluated by histological methods [103]. The levels of vascular cell adhesion protein-1 (VCAM1) and P-selectin, which facilitate leukocyte adhesion to the endothelial lining, were also decreased in lung tissue (Figure 8). Thus, KEAP1 deletion decreased the inflammation and vaso-occlusion near endothelial cells. Liver necrosis and the expression of liver damage markers were reduced in these mice.

Increased mRNA levels of *Nqo1* in peritoneal macrophages, as well as *Hmox1* in the liver, lung, kidney, and aorta confirmed that the rescue in the phenotype was due to NRF2 activation in myeloid cells [103]. In addition, myeloid-specific NRF2 upregulation promoted heme clearance and prevented its accumulation in organs. Similarly, toxic downstream metabolites of plasma heme were reduced (Figure 8). Prussian blue staining showed that iron in the liver and kidney did not accumulate abnormally upon myeloid-specific NRF2 activation. Despite these positive effects, signatures of anemia remained comparable in both genotypes (i.e., SCD::*Keap1^F/F^*::*LysM-Cre* vs. SCD::*Keap1^F/F^*), and no change was observed in red blood cell (RBC) indices, RBC lifespan or reticulocyte counts [103].

Similar to the results for myeloid-specific NRF2 activation, when KEAP1 was specifically deleted in endothelial cells (SCD::*Keap1^F/F^*::*Tie1-Cre*), the degree of liver damage, heme accumulation, and inflammation was significantly reduced. In addition, NRF2 activation in endothelial cells altered vascular leakage, which was verified by the intravenous administration of Evans blue. Compared with that in the SCD::*Keap1^F/F^*::*Tie1-Cre* mice, increased leakage of Evans blue was observed in both the liver and lung of the control SCD mice (SCD::*Keap1^F/F^*). This outcome demonstrates that NRF2 upregulation prevents vascular pathology in SCD mice by attenuating vaso-occlusion and maintaining vascular integrity (Figure 8).

Chimeric SCD mice lacking Nrf2 expression in nonhematopoietic tissues were generated to determine the role of nonerythroid Nrf2 in SCD progression [104]. These chimeric mice developed severe intravascular hemolysis despite having Nrf2 in hematopoietic cells. In addition, these mice developed premature vascular inflammation and pulmonary edema and died younger than their donor littermates with intact Nrf2-expressing nonhematopoietic cells. These results revealed a dominant protective role of nonhematopoietic Nrf2 against damage to both erythroid and nonerythroid tissues in SCD [104]. This study further supports the notion that Nrf2 activation in nonhematopoietic cells is indeed important for the reversal of the SCD phenotype. Taken together, these data reinforce the idea that NRF2 activation in myeloid and endothelial cells is a distinctive response to stressors in SCD and that upregulated NRF2 expression in both cell types synergistically mitigates SCD-induced damage in the body.

Furthermore, it has been shown that loss of Nrf2 expression leads to reduced γ-globin gene expression in mouse embryos from embryonic day 13.5 to embryonic day 18.5. Adult mouse spleens in SCD::*Nrf2^+/+^* mice have been shown to harbor a higher percentage of fetal hemoglobin (HbF)-positive cells than the spleens of SCD::*Nrf2^−/−^* mice [102]. These observations suggest that the protective effect of NRF2 activation in SCD model mice may be realized through the upregulation of γ-globin gene expression in erythroid progenitor cells and that the contribution of this regulatory mechanism to the attenuation of the SCD phenotype should be examined further.

### 4.5. NRF2 Inducers Ameliorate SCD Pathology

In line with the genetic activation of NRF2, the expression of inflammatory markers and the rate of organ necrosis were decreased when SCD mice were treated with the NRF2 activator CDDO-Im [22]. Treatment of SCD mice with DMF, another NRF2 agonist, also attenuated SCD pathology [105].

SFN, a phytochemical found abundantly in cruciferous vegetables, is a mild NRF2 inducer that is suitable for use in long-term treatment. SFN administration can reverse SCD-related pathology [23]. For example, two weeks of oral dietary supplementation with SFN was enough to attenuate liver damage and decrease the free heme levels in the plasma of four-week-old young SCD mice. Importantly, even long-term (two-month) treatment did not cause saturation of this protective effect, nor did it result in any associated adverse events [23].

SFN is an interesting NRF2 activator, especially in the case of a malady, such as SCD (Figure 9). Since SFN is a phytonutrient available as a food supplement that induces mild NRF2 activation and because SCD is evident at a young age, SFN is well suited to be a long-term treatment regimen for children [23]. In fact, a phase 1 study of an SFN-containing broccoli sprout homogenate (BSH) was tested in patients with SCD, and the results showed that BSH was tolerated in the adults and did not cause deleterious side effects [106]. High expression levels of NRF2 targeting the gene *HMOX1* have been reported in cases of SFN administration. In acute illness, however, as in the case of sickle cell crisis, short-term administration of a stronger water-soluble NRF2 inducer administered intravenously is a better option (Figure 9).

## 5. Concluding Remarks

Although inhibition of NRF2 activity in NRF2-addicted cancer cells can reduce tumor growth, induction of NRF2 activity in the surrounding microenvironment cells can reduce NRF2-addicted cancer cell expansion. These observations support the view that both NRF2 inhibitors and inducers can be used as therapeutic agents to treat NRF2-addicted cancers. However, there remain many unanswered questions. For instance, which type of cancer is attenuated by which therapy; i.e., when should NRF2 activity be inhibited and when should it be promoted? To answer this question further research is needed. It is clear that in the case of MDSCs and CD8^+^ T cells, NRF2 activation is definitely beneficial. However, in the tumor itself NRF2 activation should be repressed. Similarly, there is a paucity of research on therapies utilizing approved antineoplastic drugs in combination with NRF2 inducers or NRF2 inhibitors. Research focusing on the development of tumor-specific delivery of NRF2 inhibitors is imperative, and elaborate drug delivery systems are needed to minimize systemic inhibition.

Over the past several years, many important experiments have been carried out in the field of immunotherapy. For instance, chimeric antigen receptor-T cell (CAR-T cell) therapy was approved by the FDA for treating large B-cell lymphomas and advanced leukemia, among other cancers. In this therapy, T cells are collected from a patient, genetically modified, multiplied in the laboratory and then infused back into the patient. Considering the knowledge introduced herein, it is hypothesized that increasing NRF2 activity in these T cells before infusing them into the patient may lead to better therapeutic outcomes.

In SCD, NRF2 activation in various types of cells yields a beneficial outcome. The use of NRF2 inducers appears to be a promising therapeutic strategy for the treatment of SCD and other inflammatory diseases. In particular, combinatorial therapies similar to those used to treat cancer, are promising, but the efficacy of NRF2 inducers combined with known SCD drugs, such as HU, needs to be determined through further research. We propose that understanding the cell-type-specific roles of the KEAP1–NRF2 system will open new avenues for advanced therapies for cancer and inflammatory diseases.

## Figures and Tables

**Figure 1 antioxidants-11-00538-f001:**
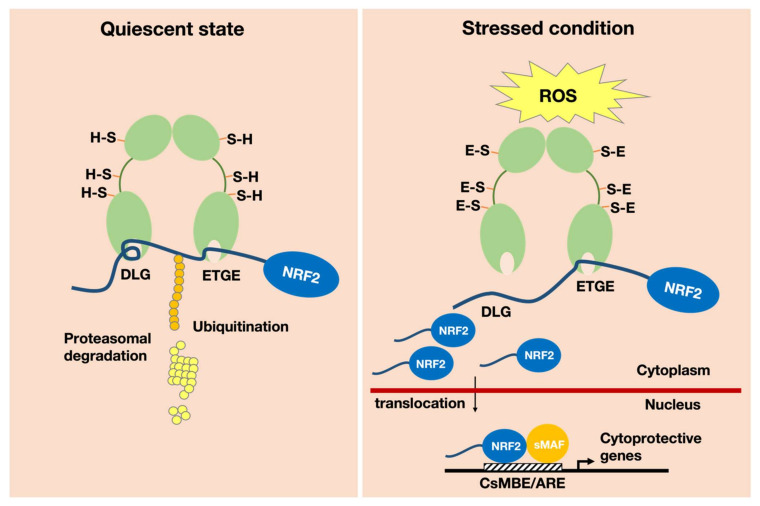
Structural and functional basis of the KEAP1–NRF2 system. The transcription factor NRF2 is a master regulator of the response to oxidative and electrophilic stresses in the human body. Under quiescent conditions, NRF2 is targeted for ubiquitination by the principal negative regulator KEAP1 (left). KEAP1 is a cysteine-thiol-enriched molecule that is the stress sensor and ubiquitin ligase of NRF2. Reactive oxygen species (ROS) and electrophiles induce modification of the cysteine-thiol moieties on KEAP1, facilitating the escape of NRF2 from ubiquitination. NRF2, thus, accumulates in the cytoplasm, leading to its translocation into the nucleus. NRF2 forms a heterodimer with the sMAF molecule and binds to a CNC–sMaf binding elements (CsMBE) or antioxidant-responsive element (ARE) motif, resulting in the transcription of a battery of antioxidant, cytoprotective, metabolizing and anti-inflammatory enzymes (right).

**Figure 2 antioxidants-11-00538-f002:**
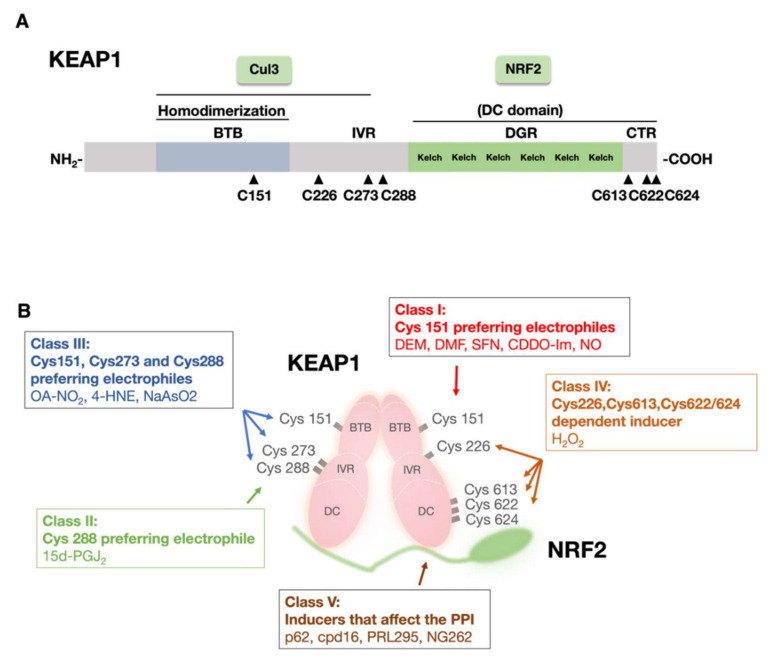
(**A**) Domain structure and function of KEAP1. KEAP1 comprises three main domains: the Bric-a-Bric (BTB) domain, intervening region (IVR), and double glycine repeat (DGR) region. The BTB domain is critical for KEAP1 homodimerization. The DC domain consists of the DGR and carboxyl-terminal region (CTR) and is important for the binding of KEAP1 to NRF2. (**B**) KEAP1 is a sensor for electrophiles and reactive oxygen species (ROS)**.** NRF2 inducers can be divided into four categories based on their preference for Cys151, Cys273, Cys288 or the Cys226, Cys613, or Cys622/624 residues in KEAP1. The fifth class of NRF2 is independent of these cysteines and activates NRF2 by inhibiting the protein–protein interaction (PPI) of KEAP1 with NRF2.

**Figure 3 antioxidants-11-00538-f003:**
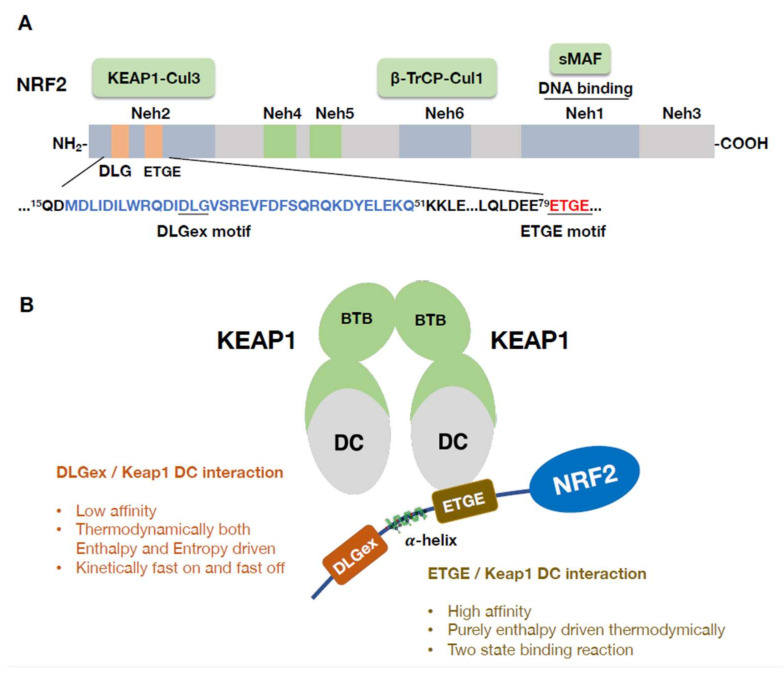
(**A**) Domain structure of NRF2. NRF2 is made up of six functional domains: Neh1-Neh6. The Neh2 domain has DLG and ETGE motifs that bind to two different portions of the DC domain of KEAP1. The Neh1 domain is important for DNA binding because it forms a heterodimer with sMAF. Neh4 and Neh5 are transactivation domains. The Neh6 domain binds to the β-TrCP-CUL1 complex to mediate KEAP1-independent ubiquitination of NRF2. (**B**) The ETGE motif has a higher affinity than the DLGex motif for binding to the DC domain. The ETGE-KEAP1 DC interaction is much stronger than that of DLGex-KEAP1 DC interaction. ETGE-KEAP1 DC binding is exclusively enthalpy driven and has been shown to elicit a two-state binding reaction. This difference in binding affinity forms the basis of the binding/hinge and latch model of the KEAP1–NRF2 interaction.

**Figure 4 antioxidants-11-00538-f004:**
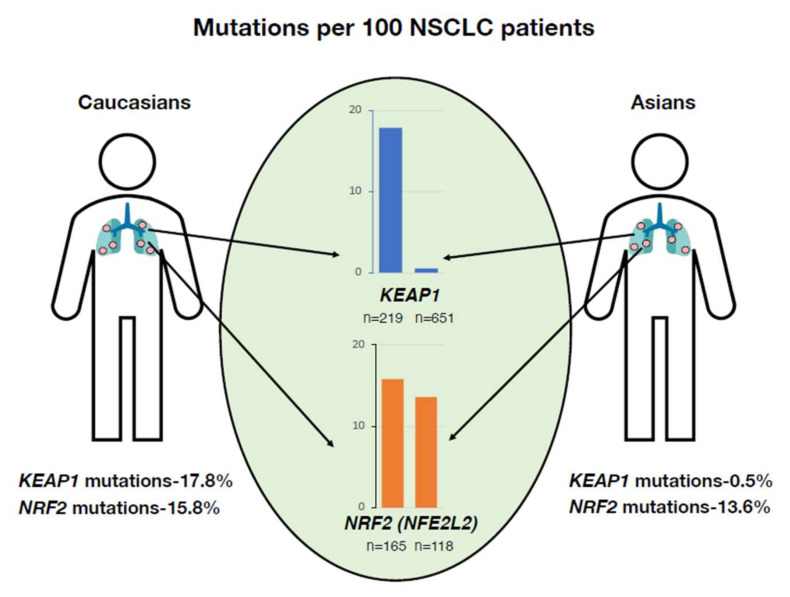
Ancestry-based disparities in KEAP1 mutations associated with non-small-cell lung carcinoma (NSCLC). *KEAP1* mutations in NSCLC are more frequently found in individuals with European ancestry than in those with East Asian ancestry. This discrepancy is not found for the *NRF2* gene (*NFE2L2*).

**Figure 5 antioxidants-11-00538-f005:**
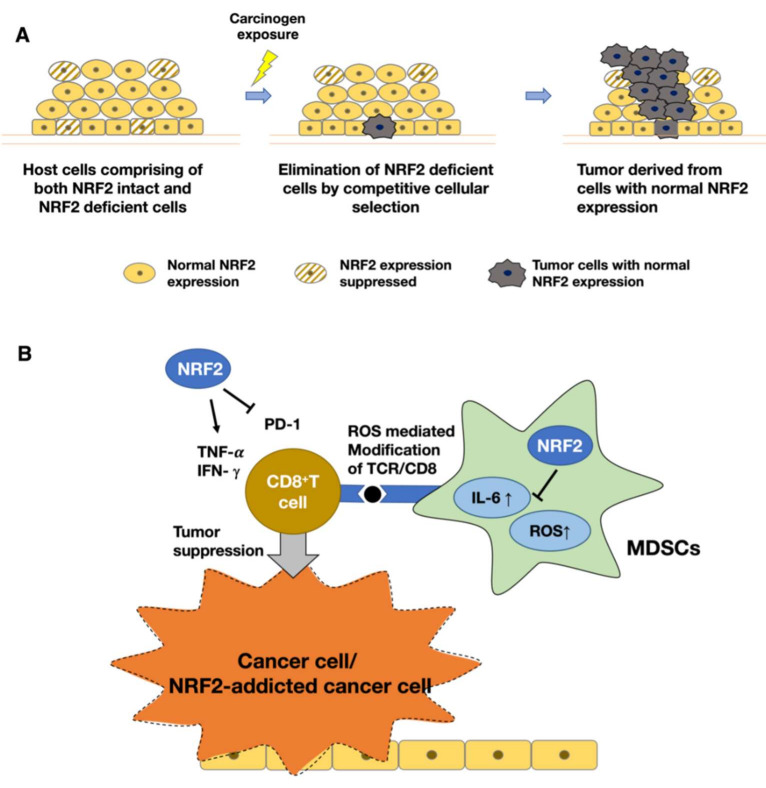
(**A**) NRF2-null cells are eliminated by competitive cell selection during chemical carcinogenesis. Upon exposure to chemical carcinogens, NRF2-null cells are eliminated by competitive cell selection, and then, NRF2-intact neighboring cells undergo carcinogenesis. (**B**) NRF2 activation in immune cells leads to tumor suppression. NRF2 activation elicits a cancer suppressive effect by decreasing the reactive oxygen species (ROS) and IL-6 levels in myeloid-derived suppressor cells (MDSCs). An increase in ROS levels modifies the interaction between a major histocompatibility complex (MHC) class I molecule and the T cell receptor (TCR)–CD8 complex. This results in a severe decrease in CD8^+^ T cell levels. NRF2 activation in the microenvironment restores the activity of CD8^+^ T cells, which also facilitates the suppression of NRF2-activated cancer. NRF2 upregulates the expression of tumor-suppressing genes such as TNF-α and IFN-γ, but it downregulates the expression of tumor-promoting genes such as PD-1 in CD8^+^ T cells.

**Figure 6 antioxidants-11-00538-f006:**
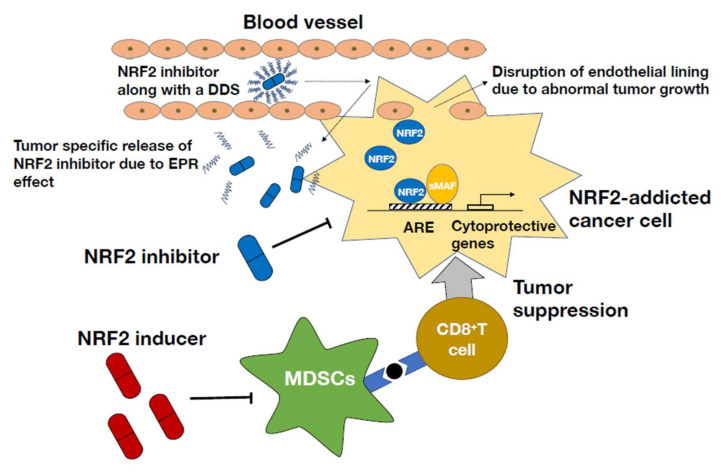
Therapeutic strategies to control NRF2-addicted cancers. NRF2 activation in myeloid-derived suppressor cells (MDSCs) leads to a decrease in reactive oxygen species (ROS) and IL-6 levels and contributes to the maintenance of CD8^+^ T cell activity. Hence, using NRF2 inducers in the tumor microenvironment is a rational approach to control NRF2-addicted cancer. In contrast, in the tumor itself, NRF2 inhibitors help suppress tumor growth and metastasis by decreasing NRF2 protein levels. However, problems posed by systemic NRF2 inhibition can be prevented by using NRF2 inhibitors in an elaborate drug delivery system designed to achieve high cancer tissue specificity through the enhanced permeability and retention (EPR) effect.

**Figure 7 antioxidants-11-00538-f007:**
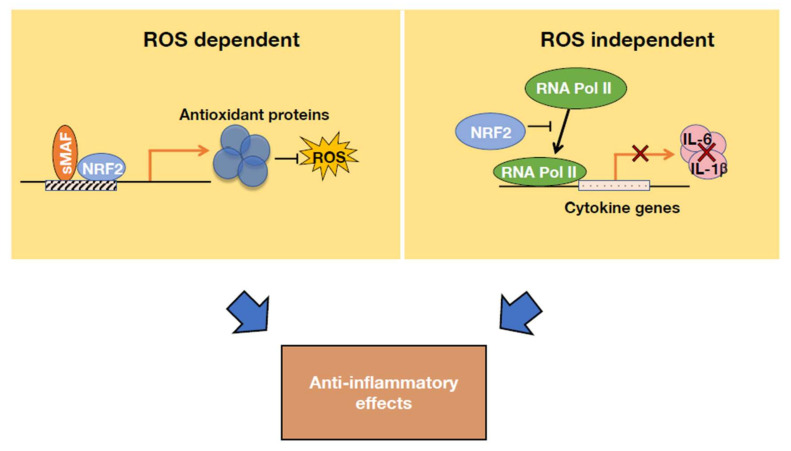
Molecular basis of NRF2 anti-inflammatory activity. NRF2 alleviates inflammation through two diverse mechanisms. First, by inducing the expression of various antioxidant proteins, NRF2 decreases the reactive oxygen species (ROS) levels in cells. Second, NRF2 directly inhibits the transcription of cytokine genes by impeding RNA Pol II recruitment.

**Figure 8 antioxidants-11-00538-f008:**
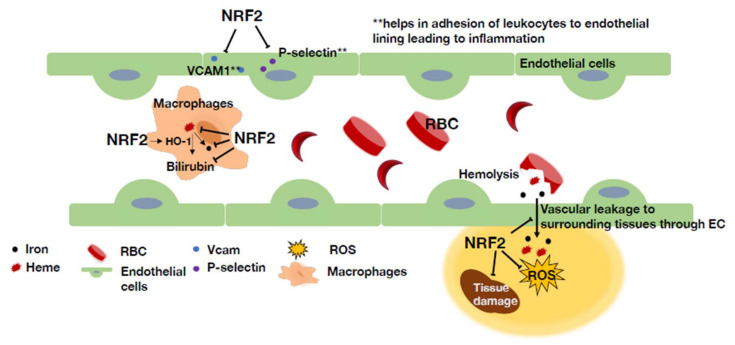
NRF2 plays multifarious roles in the sickle cell disease. NRF2 activation in macrophages contributes to increased HO-1 levels. This increase in HO-1 leads to the breakdown of hemolysis-induced release of free heme into less toxic byproducts. Increased NRF2 levels in endothelial cells help maintain vascular integrity and prevent vascular leakage into surrounding tissues. In addition, P-selectin and VCAM1 levels are also regulated by NRF2 in endothelial cells. This change in P-selectin and VCAM1 levels reduces the adhesion of leukocytes to the endothelial lining, decreasing inflammation and vaso-occlusion.

**Figure 9 antioxidants-11-00538-f009:**
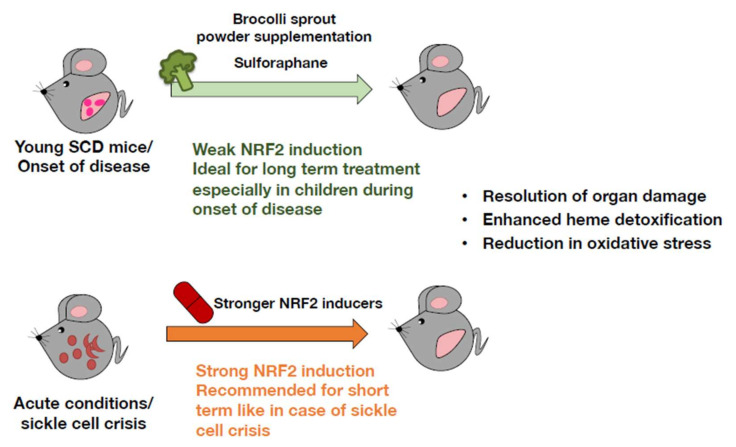
Use of NRF2 inducers as a treatment strategy for ameliorating sickle cell disease (SCD) pathology. Sulforaphane is found abundantly in cruciferous vegetables and is available as a food supplement. Because it is a mild NRF2 activator, it can be administered for a long time without causing toxicity. Therefore, it is an ideal treatment, particularly for young SCD patients upon disease onset. However, during acute conditions such as sickle cell crisis, a stronger NRF2 inducer is recommended to induce high NRF2 activation.

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
