# Peer review of "Multifaceted Roles of the KEAP1–NRF2 System in Cancer and Inflammatory Disease Milieu"

_antioxidants, 2022, doi:10.3390/antiox11030538_

Round 1

Reviewer 1 Report

The current review on the role of Keap-nrf2 system in cancer and other non-neoplastic diseases by Panda et. al. is a comprehensive, well-organized and well written review.

This article provides an up to date review of the current knowledge on the keap-nrf2 system, its mechanism of action, regulation and role in cancer and other diseases and is illustrated with helpful schematics.

Minor points:

Line 17 - Nrf2 is lower case whereas it is NRF2 everywhere else.

Addition of a glossary might be helpful for the acronyms and uncommon words (such as degron) for the reader who is not an expert in the field.

Author Response

We thank the reviewer for the comments. We have changed NRF2 in line 17 to uppercase. In addition, we have also prepared a glossary with relevant acronyms and uncommon words and placed it at the end of the text.

Reviewer 2 Report

The authors summarized a comprehensive work describing the role of KEAP1-NRF2 in cancer. The review is well presented and broadly describes many aspects of the axis KEAP1-NRF2. My comments are below:

  1. It will be good to generalize the concept to several countries.
  2. Some figures can be combined as panels in one figure. The review will benefit of a table and combination of figures, especially Figures 5 and 6 can be combined since there is a paucity of info.
  3. Figure 8 can be incorporated in the text due to lack of details and description.
  4. The authors could explain better the inhibition versus activation of NRF2. It remains an open question in the conclusion but it could be better developed in the text.

Author Response

We thank the reviewer for the constructive comments.

  1. It will be good to generalize the concept of several countries.

We surmise that this comment is regarding Figure 4 and text in section 3.1. Most of the recent publications in this context (ref. 63) has discussed this disparity amongst these two ancestries (Caucasians and Asians). That is why this section compares only these two ancestries. We went through various publicly available non-small cell lung cancer cohort data through cBioPortal, but the number of studies conducted was less and these studies did not have much interesting findings like the one mentioned in the text. Moreover, the cohort size was small in most of the studies, for instance one of the studies carried out by University of Turin, Italy, had just 41 participants. So, we decided not to include these data in this manuscript.

  1. Some figures can be combined as panels in our figure. The review will benefit of a table and combination of figures, especially Figures 5 and 6 can be combined since there is a paucity of info.

We thank the reviewer for this professional comment. We agree with this comment and merged Figures 5 and 6.

  1. Figure 8 can be incorporated in the text due to lack of details and description.

We apologize for the lack of detailed explanation related to Figure 8. We would like to keep this figure and have added several more lines to the text to explain this figure properly. We feel that this is a seminal finding and very important to understand the mechanism by which NRF2 tackles inflammation before moving into the following sections.

  1. The authors could explain better the inhibition versus activation of NRF2. It remains an open question in the conclusion, but it could be better developed in the text.

We agree with the reviewer that in the conclusion section the question whether NRF2 should be activated or inhibited has been left open in case of cancer. In case of inflammatory disease there is no doubt NRF2 activation is definitely useful and that has been clearly mentioned.

In case of cancer, NRF2 should be activated or inhibited depending on the cell type. In section 3.2.2, we have clearly stated that in case of MDSCs and CD8+ T cells, NRF2 activation is definitely beneficial. However, in the tumor itself NRF2 activation should be repressed (section 3.3). Moreover, using a NRF2 inhibitor with a drug delivery system to selectively decrease the NRF2 activity in the tumor is a good approach to avoid systemic Nrf2 inhibition and to leave immune cells with normal NRF2 activity. We have made it more clear in the conclusion section as well.

Reviewer 3 Report

The authors have been working in the field for some time. Here they summarize their own and other results and systematize the knowledge about KEAP1-NRF2. It is nice to read but novelty is low.

Author Response

We thank the reviewer for the professional comments. Regarding novelty, we would like to point out that in this review we have tried to point out “cell-specific” roles of NRF2 in the tumor microenvironment and SCD milieu, which to our best knowledge has not been discussed before in the literature.